# Mental Health Literacy Content for Children of Parents with a Mental Illness: Thematic Analysis of a Literature Review

**DOI:** 10.3390/brainsci7110141

**Published:** 2017-10-26

**Authors:** Joanne Riebschleger, Christine Grové, Daniel Cavanaugh, Shane Costello

**Affiliations:** 1School of Social Work, Michigan State University, 655 Auditorium Rd., East Lansing, MI 48824-1118, USA; cavana63@msu.edu; 2Faculty of Education, Monash University, 57 Scenic Blvd., Clayton, VIC 3800, Australia; christine.grove@monash.edu (C.G.); shane.costello@monash.edu (S.C.)

**Keywords:** mental illness, mental health, parents, children, prevention, mental health literacy, behavioral health

## Abstract

Millions of children have a parent with a mental illness (COPMI). These children are at higher risk of acquiring behavioural, developmental and emotional difficulties. Most children, including COPMI, have low levels of mental health literacy (MHL), meaning they do not have accurate, non-stigmatized information. There is limited knowledge about what kind of MHL content should be delivered to children. The aim of this exploratory study is to identify the knowledge content needed for general population children and COPMI to increase their MHL. A second aim is to explore content for emerging children’s MHL scales. Researchers created and analyzed a literature review database. Thematic analysis yielded five main mental health knowledge themes for children: (1) attaining an overview of mental illness and recovery; (2) reducing mental health stigma; (3) building developmental resiliencies; (4) increasing help-seeking capacities; and (5) identifying risk factors for mental illness. COPMI appeared to need the same kind of MHL knowledge content, but with extra family-contextual content such as dealing with stigma experiences, managing stress, and communicating about parental mental illness. There is a need for MHL programs, validated scales, and research on what works for prevention and early intervention with COPMI children.

## 1. Introduction

There are millions of children that have a parent with a mental illness. For example, between 21% to 23% of young Australians have a parent with a mental illness [1]. England and Sim [2] explored several large health care databases to find that over 7.5 million American parents had a diagnosis of major depression. These parents had over 15 million minor children. Over 12.1% of Canadian children under 12 years of age are reported to live in households with an adult diagnosed with one or more mood, anxiety, or substance abuse disorders; one of ten children live with a parent with a mental illness [3]. Five to ten percent of British non-elderly adults are parents with mental health challenges [4]. Aldridge and Becker [5] claim that up to 30% of young carers in Great Britain have parents with mental health problems.

In this study, mental illness is identified as a disorder such as major depression, bipolar disorder, and schizophrenia [6]. Children of a parent with a mental illness (COPMI) are at higher risk of developing a mental health issue, compared to same age peers [7]. Given the prevalence and risks for these children, it is important that there are appropriate and accessible supports and interventions available to them. Most of these interventions aim to provide mental health literacy (MHL) and knowledge about their parents’ illness to young people. Even though there are interventions that include educational information about parental mental health, few validated scales measure the effectiveness or efficacy of MHL in prevention interventions for children of parent with a mental illness. Indeed, there appears to be little consensus about what types of questions should be included in such outcome measures.

### 1.1. Risk and Resilience

Risk and resilience theories anchor the study reported herein. Early seminal studies conducted by Rutter and colleagues provided the groundwork for the later development of a risk and resiliency theory specifically to explain the needs of children exposed to parental psychiatric disabilities [8,9,10]. The core assumption of risk and resiliency theory can be summarized as such: exposure to high levels of biopsychosocial childhood adversity may lead to the development of future mental health difficulties [9,10]. Examples of this adversity include poor relationships in the social environment (with parents and/or peers), neighborhood violence, and exposure to parental disabilities [9]. However, there are also resiliency promoting, ameliorating factors that may reduce the likelihood that the child will experience poor developmental outcomes or future psychopathology. These include the presence of supportive relationships, coping skills, positive relationships between parents, and higher socio-economic status. Resiliency comes from learning how to handle risk with appropriate support or coping skills. Children exposed to risk in small doses may become better adjusted to dealing with future adversity [9]. However, when risk is overwhelming and omnipresent, the developing child is unable to build resistance and may face increased mental health and behavioral difficulties. These factors of risk and resiliency are composed of real, observable, events and processes that should be studied and operationalized for the creation of future interventions and prevention programs that can improve child developmental outcomes. Increasing resiliency through improved coping, elevated self-esteem, and innoculative exposure to tolerable levels of risk will help children navigate future challenges and adversity [9,11]. Risk and resilience provide further clarity by conceptualizing by considering not just specific negative or positive life occurrences, but the ongoing developmental processes that can be encouraged throughout childhood [9]. Knowledge content can be further operationalized and encouraged to promote improved developmental outcomes for children through prevention programs [12,13].

### 1.2. Children of a Parent with a Mental Illness

Children can be strongly impacted by parental mental illness. They are at a higher risk of developing behavioural, developmental, and emotional difficulties, compared to their same age peers [14]. Specific outcomes include increased dropout rates at school [15], a higher likelihood of being taken into foster or kinship care [16], and potentially developing a substance abuse disorder [17]. In addition, children of parents with a mental illness may enter out of home care if their parent is hospitalised and/or seriously unwell [18]. Children said they were sensitive to their parents ‘good’ and ‘bad’ days, at times reacting to parental behaviours with shame and embarrassment [19]. Similarly, Riebschleger [20] described the results from individual interviews and a focus group of children of a parent with a mental illness’ experiences with their parents’ good and bad days of mental health. Children defined good days as times of increased interaction, communication, and parental work and chore completion [20]. Bad days were described as times where parents were less attentive, grumpier, or more likely to yell [20].

Children of parents with a mental illness usually do not receive enough information about their parent’s mental health [19,21]. They and are left ‘guessing’ or ‘figuring out’ what is happening to their parent. Children may develop misconceptions about mental illness such as blaming themselves for their parent’s illness, or believing the mental illness can be caught like a cold and can be passed unto friends [18,22]. Adverse outcomes from a lack of open discussion about mental illness have also been described in the literature [23]. Keeping secrets from children is thought to lead to confusion for them [23]. Topics related to the illness may be ‘off limits’ for children to discuss so that there is limited or no parent-child communication about mental illness. Additionally, children may struggle to understand or worry about the parent’s behaviors or they may disengage, perhaps leading their parents to believe that they are not aware of the illness [23]. Children described how a lack of mental health information caused misunderstandings and misattributions about the causes and reasons for hospital treatment and parental behavior [19,24]. Riebschleger [20] reported that children tended to describe mental health disorders by their observations of parents’ behaviors. Only some children were able to provide the mental illness diagnostic label or had been specifically told about their parent’s mental illness [20].

Children described increased feelings of worry when they had limited information about their parent’s mental illness [25]. Without access to accurate, non-stigmatized mental health information, many children seemed to look for their own understanding of parental behaviours, often leading to confusion and distress [26]. Interviews with 20 children aged 8–22 years in Norway suggested most children were acutely aware of how their parent’s behaviour, which, at times, the children associated with feeling embarrassed, ashamed, or left out [27].

Giving a child insight and knowledge into mental illness and their parent’s difficulties is a key resilience factor within the literature [28]. Mental health literacy knowledge, especially when combined with social support, may serve as a protective factor for children who lack information and understanding of their parent’s mental illness. For example, the mental health knowledge may help children interpret their parent’s behaviour as stemming from a brain-based physical health illness that can get better [13].

### 1.3. Mental Health Literacy

Mental health literacy has been studied in the adult mental health literature for several decades. Mental Health Literacy (MHL) is defined as one’s level of understanding about mental health attitudes and conditions, as well as one’s ability to prevent, recognize, and cope with these conditions [29,30]. There are numerous mental health literacy programs, sometimes called psychoeducational programs, that aim to increase awareness of mental health challenges and facts for the general public, community crisis responders, college students, family members of adult mental health services users, and people that participate in services at mental health services agencies [30,31,32,33,34,35].

Anderson and Pierce [36] note that mental health literacy programs for adults focus on increasing mental health knowledge, confidence about helping others experiencing mental illness symptoms, as well as decreasing participants’ stigmatizing attitudes toward people with mental illness. O’Connor and Casey [34] developed mental health literacy scale items for adult mental health literacy program outcomes; they included criteria related to the recognition of mental health disorders. They also included knowledge of help-seeking, mental health risk factors/causes, self-treatments (such as coping behaviors), professional help resources, and how to get help when one recognizes mental health symptoms [34].

There is some demonstrated efficacy in the use of mental health literacy (MHL) in mental health prevention and early intervention programs designed for children of parents with mental illness [28,37,38]. For example, Solantaus et al. [39] suggest that family conversations about mental health literacy can serve as preventive interventions for children of parents with depression. Kelly et al. [40] call mental health literacy interventions for adolescents “a strategy to facilitate early intervention for mental disorders”.

Currently, there are a number of emerging interventions that aim to promote mental health literacy and wellbeing for children of parents with a mental illness. These include peer support groups [41], parenting or family-focused programs that focus on the parent-child connection [39], online support programs [42], school-based mental health literacy interventions [43], and bibliotherapy, i.e., including psycho-educational materials such as a DVD [44]. Canadian schools have demonstrated success in the implementation of school wide psychoeducational programs to improve the mental health literacy of their student bodies [45,46]. Programs focusing on parenting can be strong interventions to promote recovery and well-being in children and families that encounter parental mental illness [47].

These interventions are set within the much broader context of mental health promotion. Mental health promotion programs target a specific mental health challenges and populations at risk for acquiring these challenges. They use positive-psychology, strengths-based approaches to increase individual and community resilience and well-being [48]. A collaborative research group, with support from the World Health Organization, found strong evidence that mental health promotion and prevention programs are effective in enhancing public policy, creating supportive environments, strengthening community action, reorienting health services to include illness prevention, and developing personal skills such as resilience, social competence, and coping among children at risk for mental illness [49]. Interventions promoting children’s protective and resiliency factors aim to influence positive developmental outcomes for at risk children [12,50]. Given these findings as well as the prevalence of and associated risks for children of parents with a mental illness, it is important that interventions are developed to prevent and reduce the transmission of mental illness in families.

A review by Reupert et al. [51] of the programs for children of a parent with a mental illness described interventions, including: family-intervention programs, peer-support programs, online interventions, and biblio-therapy. A key ingredient of these programs was psycho-education. However, well-developed and tested children’s mental health literacy scales appear scarce [52]. The lack of valid and reliable child mental health literacy quantitative measures undermines the development and evaluation of evidence-based prevention programs for children of parents with a mental illness. Mental health interventions vary in their aims, evidence bases, and how they were developed. This could mean that the children of a parent with a mental illness may be less likely to receive services since funders often require programs they support to be included among databases of evidence-based practices. Resiliency is also measured using specific scales, personality inventories, and mental health literacy scales [34,53]. Thus, it is reasonable to consider that researchers compare children’s levels of risk and resiliency traits with these scales before and after interventions to assess for intervention outcomes.

Clearly, scales with robust psychometric properties are a critical and often under-resourced aspect of prevention program development and evaluation in programs for children of parents with a mental illness. Strong children’s mental health literacy measures are needed to help build evidence-based programs for children’s mental health literacy. Therefore, this thematic analysis literature review may demonstrate a step towards addressing this gap in scale construction. This paper aims to identify mental health literacy content that may be useful to inform the development of scales to evaluate the efficacy of mental health literacy mental interventions for children of parents with a mental illness. The work is guided by a research question that asks, “What does the literature indicate are children’s specific mental health knowledge needs about their parent’s mental illness?”

## 2. Materials and Methods

The study approach combined a literature review with qualitative thematic analysis, sometimes called generic qualitative analysis [54]. Qualitative thematic analysis is a flexible research method to identify ideas and patterns presented within qualitative data [55]. It is used when other forms of qualitative methods are not a good fit for the proposed work. For example, the research may not lend itself to the “prolonged engagement” needed for phenomenological methods [54,56]. This study is particularly aligned with the work of Onwuegbuzie et al. [57], who recommended combining literature reviews with qualitative methods, including theme analyses.

### 2.1. Procedures

We began by defining the problem as a need to identify the themes and content that comprise children’s mental health literacy. Our theoretical framework held that mental health literacy information may decrease risk and increase resilience for children of a parent with a mental illness. We used a number of literature sources to generate search keywords. These were drawn from university e-journal library database inquiries within ProQuest, Medline and Google Scholar. Keywords were also developed from words found in written responses of COPMI experts to a survey distributed previously by the authors herein. Key words and phrases were sometimes combined to elicit mental health literacy information specific to children, and especially children of parents with a mental illness. Keywords included: “parent mental illness”, “children (or child) of a parent with mental illness” and “child mental health literacy, knowledge, or information”. These phrases were combined with additional keywords including parent, family, stigma, resiliency, research, program, instrument, scale and/or measure.

Inclusion criteria included English language full text peer-reviewed journal articles focused on children under age 18 with some content about the mental health knowledge needs of children. We did include general child population mental health literacy program and/or scale articles because the content would include delivery to children of parents with a mental illness within schools and community programs. However, we especially looked to include the literature-identified needs and resources for children of parents with a mental illness. We accepted some articles pertaining to children of parents with mental illness knowledge needs; prevention and early intervention program evaluations for children of parents with a mental illness; and emerging child mental health literacy scales.

We excluded duplicate articles, non-English articles, those that focused on adults (18 years and older), and adult knowledge needs, programs, and MHL scales. We excluded articles that did not target mental health specifically. For example, a health literacy article with authors’ recommending teaching children about juvenile diabetes showed up in the search.

The initial pool of 238 articles was reduced to 61 after the inclusion and exclusion criteria were applied. The reasons for exclusion were that the articles focused on adult mental health consumers/patients (*n* = 61), child mental health problems (*n* = 35), services system collaboration needs (*n* = 20), only physical health (*n* = 24), and workforce preparation (*n* = 11). In addition, articles were excluded when they were repeated articles (*n* = 22), were not peer reviewed (*n* = 3), or for other reasons (*n* = 3). After initial selection of the articles, two more were excluded for having research more pertinent to adults, yielding a final number of 59 articles.

Two or more investigators read each article and independently completed a research study questionnaire form developed by the research team. The research study questionnaire was consistent with the recommendations of Beach et al. [58] for health care educational intervention literature reviews. Items on the questionnaire form included the article source (location, type and description), study methodology (study design, sampling, data collection, data analyses, limitations), and study findings. Data were collected on the sources’ recommendations of the kinds of knowledge constructs needed by children of a parent with a mental illness and how that information should be provided, i.e., by whom, what content, when to talk to the children, where to talk to the children, and how to communicate the content to children.

The thematic data analysis was drawn from the review form content with article descriptions of the problem, methods, findings, and recommendations. When a study included any kind of measurement of children of parents with a mental illness mental health knowledge, the investigators captured indepth information about the scale content, such as the name, type of instrument, description, origin, use, norming sample, psychometrics and especially the specific constructs measured with the instrument. The final database file comprised 125 pages of text.

### 2.2. Data Analysis

Thematic analysis guided the data analysis process [55]. In regular team conference calls, each article was discussed and compared as one data source with particular exploration of the child mental health literacy content ideas emerging from the data. Decision making was by discussion and consensus. Notes were prepared from team calls and became part of the data triangulation process [56].

After many reviews and discussions, each of the researchers independently developed their “top” five to seven themes. These were inserted in a shared Dropbox folder. After reviewing the folder content, the investigators collaborated to discuss the development of each theme. There were 11 main themes initially developed by the investigators. Two were the same idea portrayed with different words and were thus condensed to one agreed-upon theme. Four of the themes were included as subthemes within the operational definition of main themes. Their constructs were integrated into the operational definition of the final main theme. One was discarded by team agreement because it was more adult-focused.

Operationalized theme definitions and early coding rules began to emerge within research team discussions [59]. For example, the team discovered the need for a shared view of recovery so as to code similarly. At times, the group explored the cultural meanings of particular words that could different implications across American, European, Canadian and Australian contexts. The group also worked through discipline specific differences in orientation and training, as the team was comprised of social workers, psychologists, and behavioral health experts. One idea they agreed to accept was a consumer and family member orientation to mental illness and holistic recovery consisting of not only medical model medication plus counseling, but also mental health consumer empowerment, nutrition, exercise, work, hobbies, relaxation exercises, and social support. This was a broader conceptualisation of recovery for some members of the team but is consistent with consumer- and family-oriented recovery models of care [60,61].

Two experienced qualitative investigators independently coded, and then compared, their responses to the completed article questionnaires. They examined the extent that the data aligned, or did not align, with one or more the five main themes. The process identified further main theme construct definition criteria and coding rules. For example, a sentence on program evaluations for children of parents with a mental illness listed mental health stigma and coping in the same excerpt but since the article was primarily devoted to mental health stigma reduction, this became the primary theme. Similarly and systematically, the coders determined the content sorting into main themes for all of the 59 articles in the full database. The data were discussed, clarified, recorded and added to the operationalized definitions of the themes and their coding rules. For example, the conceptualisation of mental illness stigma was comprised of negative assumptions about people with a mental illness based only on their mental illness diagnoses, and was expanded to include anti-mental illness “attitudes” and social distance. These enhanced definitions and coding rules created a project codebook that listed the coding rules and definitions for each theme identified [56].

The researchers used another method to increase the trustworthiness of qualitative data. They completed inter-rater reliability rates by independently rating and then comparing excerpts of text for the extent of agreement with particular themes. Development of all five themes was subject to this process. Identification of the main theme aligning with particular text excerpts yielded a 90% inter-rater reliability level. Identification of other possible themes yielded a 75% inter-rater agreement level.

## 3. Results

The results are reported by a general description of the mental health knowledge themes drawn from the articles in the database. It took considerable searching to yield articles pertaining to the knowledge needs of children of a parent with a mental illness. Current knowledge of children’s needs for information and support seems to be much less developed than knowledge for parental mental illness. For example, a Google Scholar search for “parental mental illness” yielded over 668,000 articles compared to 112 articles generated for “children of a parent with a mental illness”. While there were, at times, some overlap, the articles tended to fall into one of three article focus categories, i.e., children’s needs assessments (*n* = 29), children’s program evaluations (*n* = 24), and evolving children’s mental health knowledge scales (*n* = 6).

### 3.1. Children’s Mental Health Knowledge Needs

Needs assessment article authors purported that children, especially children of a parent with a mental illness, need key mental health literature information. Three articles were policy summaries focused on knowledge and support for children and families as means of preventing the onset of mental health disorders among children of a parent with a mental illness [62,63,64]. Five sources used COPMI group and individual interviews to gather qualitative data about children’s experiences of living with a parent with a mental illness and their specific needs for information and support [13,65,66,67,68]. Several discussed how it important it is to appreciate the strengths and capacities of children of a parent with a mental illness [69,70]. Two articles examined children’s needs for parental attachment [71,72]. Several used population surveys to assess children of a parent with a mental illness’ risk of mental illness reported as elevated or “persistently high” [35,73].

Some of the information needs of children of parents with a mental illness included: to learn what a mental illness is [19], the different types of mental illness [68]; the causes of mental illness [74]; finding out if the parent is likely to get better or prognosis [14]; learning how to cope with the parental illness symptom fluxes [14]; and where to seek help and support [25]. Children also said they could benefit from learning what goes on in mental health services, and how to communicate with others about their parent’s illness [20].

It is also important to note that exhaustive literature reviews comparing a majority of mental health literacy measures have recently been conducted [75,76]. The researchers found 89 validated measures that covered one of three mental health literacy constructs; help-seeking, stigma or mental health knowledge [77]. Most of the measures were for stigma and omitted other components of mental health literacy. Additionally, there were only four knowledge measures appropriate for children and adolescents [78]. While comprising important new knowledge development, it does not appear that any of the reviewed measures were drawn from research especially directed towards COPMI.

### 3.2. Evaluations of Programs for Children of Parents with a Mental Illness

Among the program evaluations, several were a literature review of summative and formative descriptions of current mental health literacy programs [50,79]. Most were descriptions of specific mental health knowledge enhancement programs for children in general; some were focused on children of parents with a mental illness. Among the 29 program evaluation sources, four employed intervention and wait-list control groups with pre, post, and follow up outcome assessment [12,41,45,80]. For example, Perry et al. [80] randomly assigned general population schoolchildren to a mental health psycho-educational intervention while others remained in their regular academic courses. For the most part, these studies revealed decreased mental health stigma as the main outcome for program children.

Most of the programs included mixed methods designs that included children of parents with a mental illness oral or written input about their learning in response to open-ended questions, plus the addition of some kind of quantitative measures, including program-developed questions that asked children to report their levels of awareness about mental illness and recovery [81,82]. Some used additional standardized scales to assess the participants functioning such as a depression scale, a hope scale, and/or a coping scale [13,39,66]. Four of the programs used a family approach where children participated in a talk about the parental mental illness [39,83,84], or in setting family goals [82]. In general, family communication about parental mental illness was improved.

Many of the programs were peer-focused; professionals provided mental health information to children of a parent with a mental illness [12,13,67,81,85,86,87]. One program used a DVD and a follow-up conversation to provide mental health information to children of parents with a mental illness [37]. Numerous articles described school-based interventions to teach children about mental health; many of these programs particularly focused on reducing mental illness stigma [80,88,89,90,91,92]. Finally, Jorm [93] used telephone interviews to find out the extent young Australians were aware of mental health organizations that provide supportive mental health information programs and direct services; unfortunately, many of the interviewees had little knowledge of the available programs.

Gladstone et al. [19] suggest that psycho-education programs should attempt to include young people’s views of their parent’s illness and should recognise the children’s roles within the family context. Additionally, children may want help with practical issues such as providing support when their parent is hospitalised and when they need a break from caring for their parent [18,22,74].

### 3.3. Emerging Children’s Mental Health Literacy Scales

There was significant overlap between program evaluation and emerging children’s mental health literacy scales. While 15 articles described the use of quantitative measures of mental health literacy, six of these were designed to measure program-specific learning rather than mental health literacy more broadly. Of the remaining measures of mental health literacy, several were used in only one article. Two measures appeared in two articles. The first appeared in Wahl and colleagues in 2011 and 2012 [43,91]. The second appeared in Fraser and Packenham in 2008 and 2009 [12,94]. Finally, another measure appeared in three articles [37,44,95].

A variety of question types were used in the children’s mental health literacy measures. The most common type of question was a multiple choice [13,37,44,80,89,94,95,96]. Several measures used open-ended questions, with participants receiving higher scores for more accurate responses [87,92,94,96]. Two measures used a multiple choice format, where more accurate responses were scored higher [43,91]. Two measures used vignettes to present contextual scenarios [96,97]. All measures were comparatively brief, with a median length of seven questions per measure, ranging from five to 28 items.

Generally, little detail was provided about the measures themselves. With the exception of test-retest reliability [43,91], and internal consistency [43,89], no other psychometric data were reported. One article cited test-retest reliability and internal consistency for its measure from an adult sample [80], but did not report it in the child sample. No article provided evidence of convergent and discriminant validity for the measures. There was also little detail provided regarding the content of the unpublished measures.

With the exception of two measures which focused predominantly on knowledge about depression [80,97], all instruments identified as measuring mental health and illness knowledge in the broad sense. Depression-specific measures used either vignettes or multiple choice questions, which focused on the identification of symptoms such as suicidal ideation, emotional distress, loss of interest in enjoyable activities, loss of appetite and weight loss, sleep disturbance, poor concentration, and fatigue. Examples of broad mental health literacy items included “People who have had mental illness include astronauts, presidents, and famous baseball players” and “Most people with severe forms of mental illness do not get better, even with treatment” [43]. Other areas of mental health literacy reflected in the identified measures include stereotypes and beliefs [37,44,95], as well as recovery and stigma [13,96]. An example of a recovery item was “Mental health treatment usually works as well as treatment for other health problems”, and a stigma item was “It is not easy to tell if someone has a psychiatric illness by looking at them” [13]. The paucity of detail regarding the content of items used to measure mental health literacy was notable, with most articles not including the measures in full or even providing sample items.

### 3.4. Children’s Mental Health Knowledge Content Themes

Over 30 sources recommended mental health literacy programs and initiatives for increasing children’s knowledge of mental illness and recovery. The thematic analysis yielded five main themes of mental health literacy for children of a parent with a mental illness These knowledge themes focused on children, especially children of a parent with a mental illness: (1) attaining an overview of mental illness and recovery; (2) reducing mental health stigma; (3) building developmental resiliencies; (4) increasing help-seeking capacities; and (5) identifying risk factors for mental illness.

Theme One: Attaining an overview of mental illness and recovery. Many of the authors of the database articles asserted that children, especially children of a parent with a mental illness, need to acquire an overview of mental illness and recovery. However, exactly what kind of information should be delivered was less clear. Across the sources, there were references to children of a parent with a mental illness needs for knowledge of mental illnesses including depression, anxiety, bipolar disorder, borderline personality disorder, schizophrenia, substance abuse, and co-occurring mental illness and substance abuse [20,46,84,97,98].

Some authors asked children to learn the symptoms of particular mental illnesses using psychiatric diagnoses behavioral criteria. For example, Wright et al. [99] examined the extent that children could recognize signs of depression and psychosis. Similarly, Lam [100] used mental health case vignettes to survey the extent that 1678 students were able to label depression and report their willingness to seek help. For the most part, it was unclear how the overview of mental illness content was to be adapted for ages of children and their levels of development. One exception is an early article by Henderson [101] that describes discussion questions for middle school COPMI students attending a school-based counseling group.

One area that seemed clearer is that some authors suggested children should learn that mental illness was a health condition and it could often get better with treatment and recovery [68,91,102]. Some suggested that children be taught that mental illness affects at least one in five people and is therefore a very common health condition that affects many people, as well their families [13]. Other than referring to “getting help” for mental illness, it was not always clear what children were told or should be told about mental health recovery. It did appear that most frequently, the authors spoke of medical model treatment of medication and counseling, followed by support, and sometimes including stress management, health habits, and life activities [13,68,88,99].

When compared with general population children such as those in school classrooms, COPMI-focused articles were more likely to identify the need for mental health literacy to contain content on family experiences of living with a person with a mental illness. For example, they identified family “good days and bad days” linked to the symptoms levels of the person with the mental illness [20]; and “navigating an unpredictable daily life” [103]. Children needed to be able to engage in problem solving for assessing how they should respond to parents on “good days” and “bad days” [20], particularly as some COPMI claim family connections and interactions vary with the cyclical nature of mental illness symptoms of a parent [98].

A number of articles contained content supporting the use of peer sharing among children of a parent with a mental illness’ awareness that many other children have similar family experiences [41,68,81,85,101]. Similarly, some database sources described children of a parent with a mental illness and family experiences in having parent-led family discussions about the specific illness of the parent and family member impacts, feelings, interactions, and communication [39,51]. These could also include parental crises, episodes, hospitalizations and child-parent separations [51]. Reupert et al. [51] listed specific key messages for parents talking to their children including: children don’t cause mental illness; children don’t need to “fix” the parent’s illness; the parent is getting help; it’s okay for the child to ask questions; it’s okay for the child to say how she or he feels about the illness; and there are supportive people the child can access. Additionally, Grové and colleagues [95] explained that children are not responsible for taking care of their parents; the authors emphasized the need to convey this as part of mental health literacy for children of a parent with a mental illness.

Theme two: Reducing mental health stigma. Mental health consumer parents said that their minor children received most of their knowledge about mental illness from the media; they said this led the children to have stigmatized views of people with mental illness [102]. A plethora of studies focused on mental health literacy help to reduce children’s assumptions or “attitudes” that people with mental illness are incompetent, unstable, immoral, violent and people to be avoided [92]. Instead, children can learn that most people with mental illness can work, make decisions, develop their talents, and contribute to the world [36,88]. It appears that reduction of children’s mental illness stigma is a main outcome of many mental health literacy programs; many of the programs reportedly challenge mental illness misconceptions [80,91,104]. Many of the program evaluation articles were drawn from pre- and post-mental health literacy program outcomes with sampling drawn from whole classrooms of general population children [43,89,90,91,92,105].

COPMI-focused sources identified a need to address general mental illness stigma as well, but with the addition of discussing stigma experiences of children in living in a family with a person with a mental illness [85,86]. This can include external sources of stigma such as teasing from peers and others; it can also include self-stigma such as being embarrassed by parental behaviors [98,103]. Mental health literacy for COPMI children can include helping children develop strategies for dealing with external and internal stigma experiences [98].

Theme three: Building developmental resiliencies. Children of a parent with a mental illness who learn how to deal with mental health stigma may demonstrate enhanced developmental resiliency in the face of child and family stigma experiences [37,65]. Access to mental health information was sometimes linked to the idea that “knowledge is power” for children of a parent with a mental illness [86]. Children exposed to accurate, non-stigmatized mental illness and recovery information reported they were able to understand their parents’ behaviors, share stories with peers, talk to their parents about mental illness, and they felt less alone [86]. For example, some children reported that being able to talk about parental mental illness with peers was “like a weight off your shoulders” [86]. Social support from nurturing adults, siblings, and peers helps strengthen developmental resiliencies of children of a parent with a mental illness [85,94].

It is also important for children to have meaningful connections and attachments with parents and other caregivers [71,72]. Supported parents may build more positive and stronger relationships with their children; the quality of children’s relationships with their parents is an important resiliency factor [83,98,106]. Family communication, including communication about parental mental illness, may help children of a parent with a mental illness build additional developmental resiliency [51].

Many sources described mental health literacy programs as mechanisms for prevention of mental illness for children of a parent with a mental illness [39,64,81,83]. For example, Solantaus and colleagues [39] found that children of a parent with a mental illness who participated in a family discussion about a parental mood disorder showed decreased reported emotional symptoms, anxiety, and hyperactivity, as well as increased pro-social behavior. These appear to indicate children’s enhanced resiliencies.

Some of the articles discussed the need for children of parents with a mental illness to learn and practice stress management skills toward building resilient behaviors [13]. Riebschleger and colleagues [13] reported that children need to learn how to reach out to supportive people and resources; the children may need personalized, active stress management plans [68,103]. Active stress management skills are likely to decrease risks of mental illness. In addition, it is important to recognize the strengths of children of a parent with a mental illness; for example, some children of a parent with a mental illness report that they have strong skills in crisis management, problem solving, and caregiving [69,70]. It is also important that children of a parent with a mental illness learn how and when to help, or not help, others, including parents and other family members [102].

Theme four: Increasing help-seeking capacities. Many sources identified the need for children, especially children of a parent with a mental illness, to seek help for personal distress, and sometimes, parental and family crises [78,93,97,104]. In fact, some sources described mental health literacy programs as mechanisms of early intervention should mental illness symptoms appear among the children; the logic is that children are more likely to recognize mental illness symptoms and to know how to seek help [40,63]. Children who view mental illness as health condition that is no one’s fault may be more likely to seek help [13]. If children, including children of a parent with a mental illness, have some basic information about recovery and think recovery can be effective, they may be more likely to seek help as well [78,96]. Further, stigma may be much less of a barrier to help-seeking for child participants of mental health literacy programs [46,91].

Some sources noted that help-seeking of children of a parent with a mental illness help-seeking should including an awareness of professional and community mental health treatment resources [93], as well as children of a parent with a mental illness reaching out to trusted adults, peers, siblings, and others [78]. A few sources mentioned COPMI help-seeking can be part of mental health knowledge areas of preparing for crisis management and safety planning [13,106].

In addition to increased child reported help-seeking capacities, numerous mental health literacy program evaluation sources indicated that child mental health literacy program participants reported they were more likely to engage in providing help to people with a mental illness. For example, Olsson and Kennedy [107] found adolescents exposed to vignette-based hypothetical situations about a peer with mental illness symptoms were able to say how they would respond, or help, the peer described in the case example. The ability to seek and provide help can be associated with higher levels of developmental functioning and resiliency. Yap and Jorm [108] surveyed 1520 Australian youth and found that subjects reporting less mental illness stigma attitudes also said they would be more likely to help a friend or family member with a mental health problem.

Theme five: Identifying risk factors for mental illness. Numerous sources identified the need for children to be able to identify possible mental health conditions, such as depression, by the duration and intensity of particular diagnostic behaviors within case examples [99,100,107]. The ability to identify early warning signs of mental illness were said to be desirable for promoting child/COPMI help-seeking behaviors [109]. Very few of the articles seemed to identify particular early warning signs [109], such as isolating oneself or persistent sadness. Some sources appeared to reference a combination of biological and environmental stressors as precipitators of the onset of mental illness symptoms [62,108]. A few sources suggested that children of a parent with a mental illness need information about heritability of mental illness [13,87]. Some sources said that children can explore factors that might help mitigate risk factors for mental illness, such as using active stress management, connecting with others, communicating with family members about mental illness, finding community resources, and engaging in good health habits [81,103]. Jorm [109] provided a list of strategies suggested for those with subclinical depression such as maintaining a good sleep schedule, engaging in exercise, learning relaxation, and doing something one enjoys. Despite elevated statistical risk factors, a few of the sources suggested there is a need for children of a parent with a mental illness to experience feelings of hope about their families and their futures [13,68].

## 4. Discussion

The research question asked, “What does the literature indicate are children’s specific mental health knowledge needs about their parent’s mental illness?” The main themes of the data were: (1) attaining an overview of mental illness and recovery; (2) reducing mental health stigma; (3) building developmental resiliencies; (4) increasing help-seeking capacities; and (5) identifying risk factors for mental illness. The findings revealed some general knowledge constructs for children as a whole, often set within classroom mental health literacy programs. Other sources focused on the additional knowledge needs of children of a parent with a mental illness.

Theme one data seemed to show that an overview of children’s mental illness and recovery content included identifying specific illness behaviors and symptoms, mental illness as a health challenge, and recovery actions such as taking medications, participating in counseling, managing stress, finding support, and maintaining good health habits. A main message for children seemed to be that people could get better within mental health recovery. Information especially pertinent to children of a parent with a mental illness was the need to deal with specific family member behaviors changes from day to day, as mental illness symptom levels vacillated. Mental health literacy content on problem solving and day to day coping was recommended. The need for peer sharing and parent-led family discussions about mental illness was also advised.

Theme two data indicated that children need to know about mental health stigma. This was often presented within illustrative case vignettes. Some sources found that lower levels of mental illness stigma seemed to promote increased willingness to seek help for mental illness concerns. Children who learned about stigma and may have replaced these ideas with factual information about mental illness also seemed to report they were more willing to help peers and family members that had mental health challenges. Children of a parent with a mental illness reportedly needed to process external family stigma episodes when others teased them or their family members with a mental illness. They also had to deal with internal stigma, including how they felt about themselves and their family member’s mental illness. Children of a parent with a mental illness may be able to benefit from developing strategies for responding to internal and external stigma.

Theme three reported that mental health literature revealed that is it useful for children to know how to seek help for themselves and others in the event of a mental health concern. However, theme four findings largely target children of a parent with a mental illness who may use mental illness and recovery information as a beginning step toward reducing the impact of developmental risks. For example, children could glean more knowledge of their parent’s illness. They may be more likely to begin to talk to their peers and their parents about their mental illness. There was a strong recommendation within a number of database sources for children of a parent with a mental illness to engage in outreach to supportive individuals and, especially, to create and implement active stress management planning. Information for children of a parent with a mental illness should also include identification of strengths.

Theme four data seemed to indicate that children need to be able to increase their mental health help-seeking capacities. Children of a parent with a mental illness may particularly benefit from learning that mental illness is not anyone’s fault and it is certainly not the children’s fault. They may find there are times when they may need to seek help for their own personal distress and perhaps even family or parent crises. It is also important that children of a parent with a mental illness learn how and when to help or not help others, including parents and other family members. Some sources identified a need for the children to develop crises and safety planning.

Theme five data appear to relay that children need some information about risk factors for mental illness. Some sources said children would need to be able to identify early warning signs of mental illness. However, few sources said what these early warning signs were except to say they would be tied to particular diagnoses, such as depression. A few sources said that mental health literacy information would likely identify possible biopsychosocial influences for developing mental illness. COPMI-focused sources particularly emphasized the need for coaching these children in active stress management, connecting with others, communicating with family members, finding community resources, and engaging in good health habits.

The findings for children as a general target group, such as those used in school-based mental health literacy interventions seem to align well with the work of O’Conner and Casey [34], Wahl et al. [44,91], and McLuckie et al. [89], all of whom delivered mental health literacy programs in school classrooms. These sources seemed to consider the original work of Jorm and colleagues’ original work in mental health literacy for adults [29,30,109] and with young people [93]. Specifically, they included content on a person’s abilities to: recognize specific disorders, know how to seek mental health information, identify mental illness risk factors and causes, engage in self-care, know professional help resources that are available, and seek help after recognizing emerging mental illness warning signs or symptoms. In this study, the findings also indicated the need for these constructs under themes of obtaining an overview of mental illness and recovery; decreasing mental health stigma; increasing help-seeking capacities; and identifying risk factors for mental illness.

It is noted that the thematic finding of building developmental resiliencies is consistent with some of the more COPMI-focused articles such as those by Foster et al. [66,85], Gladstone et al. [67,70], Grové et al. [37,44,95], Mordoch and Hall [25], Riebschleger [20], and Riebschleger et al. [13] These needs and program assessment sources particularly focused on the interactions between children of a parent with a mental illness and their parents, peers and others. It appears that this thematic finding may support risk and resilience as a theoretical foundation for mental health literacy programs for children with a parent with a mental illness. They seem to align with Rutter’s recommendations for resiliency-promoting, ameliorating factors such as children having supportive relationships, coping skills and learning how to deal with family-related emotional risk factors, such as stigma experiences [9].

### 4.1. Strengths and Limitations

This study has a number of limitations and strengths. A lack of detail in many of the articles sometimes made it difficult to identify the child mental health literacy content. For example, many of the articles evaluating children’s mental health literacy referred to measures or scales but they did not provide scale questions. Program evaluation sources sometimes did not prove details of the program content covered. However, among those that did provide scale questions or program description, it was possible to glean some useful data about child mental health literacy content. The use of a form for the literature reviews further reduced mental health literacy content detail available. In some cases, it was necessary to go back to the original articles for more clarification. Finally, engaging in thematic analysis using literature formatted in third person and whole study perspectives was challenging. However, as indicated by Onwuegbuzie et al. [57], it was possible to engage in a thematic analysis of a literature database.

Other limitations of the study include the early level of development of the child knowledge base of mental health literacy and measurements of child mental health literacy levels. However, given the limited number of sources for children’s mental health literacy and even less for mental health literacy for children of a parent with a mental illness, the researchers were still able to draw a good deal of children’s mental health literacy content from COPMI needs assessment, program evaluations, and emerging scales.

The data sources herein were drawn from only three databases. It is possible that other children’s mental health literacy sources were available among other databases not selected, but well-known and broad spectrum databases were used. Since only articles in English were included in the database, this remains a limitation of the study.

Qualitative studies require interpretation by research analysts, so bias can be a concern. Creswell [56] says that the trustworthiness of the data can be strengthened when the researchers use two or data collection and analysis strategies. In this study, the researchers used data triangulation, developed a code book that may be used for replicating the study, and demonstrated good inter-rater reliability for data coding.

### 4.2. Recommendations

With the early state of knowledge development in mind, it may still be possible to offer some modest recommendations for practice, policy and research. There appears to be a need for mental health literacy programs for children from the general population. Many of these are school-based and have demonstrated that they appear to increase knowledge of mental illness and to reduce mental health stigma among children. Children of parents with a mental illness need the same basic information with a good deal more additional information tied to family experiences with mental illness, including talking to family members about mental illness. They need accurate, non-stigmatized and family-contextual mental health literacy information. There is a need for children of a parent with a mental illness to learn how to build skills as well as knowledge for help-seeking, helping peers/family members, responding to stigma experiences, talking to peers, actively managing stress and feeling hopeful.

Since mental health literacy programs are needed for children, policy makers need to prioritize and fund the delivery of mental health knowledge to children and to children that have a parent with a mental illness. This enhances the development of manualized, evidence-based mental health literacy programs for all children and children of a parent with a mental illness. Workforce preparation is needed as well as collaboration between federal, state and community agencies that deliver health, mental health, child welfare and educational services [110]. The development of the programs should include the active participation of children of a parent with a mental illness and their parents. Support for parental roles should be further explored and supported. Recovery frames may need to be more holistic and less medical model in orientation. For example, the self-care suggestions of O’Connor and Casey [34], as well as Jorm [93], should be included, in addition to those recommended by children, parents, and other family members. It may be useful to consider the inclusion of strength-based and empowerment theories within program content and delivery processes.

Future research should be well funded to execute larger sampling and more rigorous designs. In fact, there is need to study what kind of mental health information should be available to children, at what age, and how it should be delivered. Evidence-based child mental health literacy programs will require the development of validated scales with strong psychometric properties. It may be necessary to develop children’s mental health knowledge scales for particular age children. COPMI children may need additional questions specifically targeted toward their family experiences, such as dealing with stigma experiences, managing stress, and communicating about parental mental illness. The scales need to align with children’s developmental levels and backgrounds. Thus, there is likely to be a need for variations of the scales by age groups, family circumstances, languages and cultural group affiliations.

## 5. Conclusions

It is clear that there is much work to do in order to ensure that all children, especially COPMI, have access to accurate mental health information and support. Mental health literacy programs could help increase accessibility of the information but they require mental health literacy scales and strong evaluation to show “what works” for building children with high levels of mental health literacy. Practice, policy, and research could help millions of children of a parent with a mental illness find increased family understanding, coping, and hope for their developmental futures.

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
