# Peer review of "Mental Health Literacy Content for Children of Parents with a Mental Illness: Thematic Analysis of a Literature Review"

_brainsci, 2017, doi:10.3390/brainsci7110141_

Round 1
Reviewer 1 Report
A very nice piece of work. Suggest find more recent references on Mental Health Literacy in schools to inform the more current understanding of the topic - Try Canadian Journal of Psychiatry; European Child and Adolescent Psychiatry; Psychiatric Clinics of NA. The format or your paper fits perfectly with more current definitions of MHL based on WHO health literacy framework.
Suggest you find the 3 recent systematic and detailed papers by Wei et al on the measurement of MHL in young people. Use this to help you address that part of the paper.
You may want to check out FAMILY PACK on teenmentalhealth.org which is a specific resource (widely available and used for at least 5 years now) that has incorporated many of the components you have discussed.
Author Response
I'm attaching a file with the authors response to reviewers' comments. Thank you for your review.

Reviewer 2 Report
The paper provides a generally solid thematic analysis of the literature review - nice work! The findings are sound and not surprising. They highlight the critical need for more focused attention on mental health literacy - for all children - including children with parents with a mental illness. And while it is critical to identify the additional needs for different types of support among COPMI, it is critical not to imply to readers that the children should be segregated. In addition, while it is understandable that the literature would focus on mental illness, very little attention is given to the large body of literature available on mental health promotion and prevention and the growing evidence regarding their positive effects on mental health and resilience. Further, while the literature review shows reduced stigma with MHL regarding mental illness, it would be very useful for the literature review to also address the effects of mental health promotion actions on stigma.
Finally, the sentence structure needs a solid review.
Author Response
I'm attaching a file that details the authors' responses to reviewers comments.

Reviewer 3 Report
Dear authors,
The manuscript offers additional information to the field of Children of Parents with Mental Illness (COPMI) in a clear overview. It gives good possibilities to develop programs for COPMI and scales to assess their needs. Resilience and recovery seem to be important principals, that started with adults and in this article, it is used for children.
The introduction contains a clear overview of literature about Children of Parents with Mental Illness and their Mental Health literacy and lack of information.
In the Results section a lot of literature is analyzed. Besides an overview of mental health knowledge in general it gives a preparation for a mental health knowledge scale for children.
Some comments:
In the abstract three themes are described:
1. Findings revealed that children’s MHL may consist …..
2. Child MHL scale constructs
3. COPMI appear to need the same kind of MHL knowledge content,
These themes have some overlap in content and some content is different. For the reader it is hard to understand the differences in the way how it is described.
144 a recent reference of a systematic review of parenting programs with recovery point of view can be added:
Reupert, A., Price-Robertson, R., & Maybery, D. (2017, February 9). Parenting as a Focus of Recovery: A Systematic Review of Current Practice. Psychiatric Rehabilitation Journal. Advance online publication. http://dx.doi.org/10.1037/prj0000240
2. Materials and methods are clearly and comprehensive described. It can be more concise.
Small corrections
18. Finding must be findings
29. This prevalence figure is from Australia. Add some data from the US and UK
54. Chidren à children
195 mental per a survey. It seems some words are lacking
296 Children’s Mental Health Knowledge Needs Assessments
309 + 771 Systema àSytema
410 Mention ‘Theme one’ here
647 paen à parent
Author Response
We attach the authors' responses to reviewer comments. You will need the updated new manuscript draft to check it. We have e-mailed the editor asking it be sent to you. It can't be loaded at this screen point.
